# Effect of N Fertilizer Dosage and Base/Topdressing Ratio on Potato Growth Characteristics and Yield

Xiaoting Fang [1,2,†], Zhuqing Xiang [2,†], Haiyan Ma [1,2], Fang Wang [3], Qiang Wang [1,2], Peihua Li [3,*] and Shunlin Zheng [1,2,4,*]

1   State Key Laboratory of Crop Gene Exploration and Utilization in Southwest China, College of Agronomy, Sichuan Agricultural University, Chengdu 611130, China
2   Crop Ecophysiology and Cultivation Key Laboratory of Sichuan Province, Chengdu 611130, China
3   Panxi Crops Research and Utilization Key Laboratory of Sichuan Province, College of Agricultural Sciences, Xichang University, Xichang 615013, China
4   Key Laboratory of Tuber Crop Genetics and Breeding, Ministry of Agriculture, Chengdu Joyson Agricultural Technology Co., Ltd., Xindu 610500, China
*   Correspondence: peihuali2000@163.com (P.L.); zhengshunlin123@163.com (S.Z.)
†   These authors contributed equally to this work.

**Abstract:** Nitrogen fertilizer is an important kinetic energy for potato production. To study the effect of nitrogen(N) fertilizer application and the topdressing ratio on potato growth and yield, different N application levels: N75 (75 kg ha$^{-1}$), N150 (150 kg ha$^{-1}$), N225 (225 kg ha$^{-1}$) and different N fertilizer base/topdressing ratios: T1 (2:8), T2 (5:5), T3 (8:2), and T4 (10:0) were used to find the best N fertilizer operation. The results showed that higher yields can be obtained when 120–180 kg ha$^{-1}$ N was applied under the base/topdressing ratio of 8:2 in silt loam (sand–silt–clay: 29-50-21). The base N fertilizer dosage largely determined the root morphology, while topdressing fertilizer mainly affected the leaf photosystem, however, rhizosphere topdressing at the early stages of bud emergence did not immediately affect the leaf soluble protein and the sugar content. Topdressing N increased the tuber crude protein and ascorbic acid content to some extent, but had weak effect on the amylose/amylopectin ratio, for the starch content was mainly influenced by total N application. When base N fertilizer was low, it could be compensated by applying fertilizer during flowering, though when the amount of base fertilizer was sufficient, topdressing would cause a decrease in the average single potato weight and an increase in potato sets per plant, thereby reducing the commercial potato rate. Overall, adopting a suitable base/topdressing ratio can promote plant growth and improve tuber quality and yield. However, the relationship between the specific application amount and the ratio under different soil texture conditions needs further study.

**Keywords:** potato; N fertilizer; base/topdressing ratio; growth characteristics; yield

## 1. Introduction

Potato (*Solanum tuberosum* L.) is an important staple crop grown in more than 150 countries worldwide, with a total production of 376.1 million tons in 2021 [1]. Nitrogen (N) is an essential nutrient for potato growth and development. As early as the 20th century, scientists were concerned about the effects of nitrogen transformation on potato yield and dry matter accumulation [2]. Potato responds differently to various forms of nitrogen ($NH_4^+/NO_3^-$) [3], and the effectiveness of nitrogen in rhizosphere soil affects plant growth and development, with soil indigenous N contributing to 68.5–75.0% of tuber yield [4]. Some studies have shown that increasing the amount of N applied within a certain range (about 100 kg ha$^{-1}$) can effectively increase potato yield, while exceeding the threshold (about 200 kg ha$^{-1}$) has a negative impact, though cultivars and seasons should be critically taken into account [5,6]. N fertilizer is often applied as a onetime-based treatment in cultivation, while an application that happens proximately to planting time can improve

nitrogen utilization [7]. Therefore, topdressing came into cultivators' vision. Potatoes tend to live in shade, but the loss of N fertilizer by leaching out in winter can be as high as 50% [8]. A split application of N fertilizer not only improves N fertilizer utilization but also reduces susceptibility to early blight [9]. Nitrogen demand at the time of application determines the ability of the crop to compete for N [10]. Potatoes are also sensitive to fertilizer during reproductive growth, with maximum daily N uptake at 55–70' d of seedling emergence [11], the proportion of N fertilizer leaching is also high at this time [12]. Increasing base N fertilizer will produce continuous N stress [13], but extending the nitrogen application cycle improves soil N utilization rate and maintains a high nitrate content [14]. Meanwhile, N utilization can be improved to some extent by reducing the base fertilizer and increasing topdressing amount.

In rice, topdressing can improve the photosynthetic performance of plant leaves and seed yield [15]. In soybeans, fertilization can regulate leaf nitrate metabolism [16]. Unlike seed crops, inappropriate fertilization may result in delayed maturity and lower yields in potatoes, a crop with underground tubers harvested as the economic organ [17]. Potato production should not only focus on the accumulation of dry matter in the tubers, but also the nutritional and processing quality of tubers [18], however, different varieties of starch synthase-related genes respond to different N concentration gradients [19]. Nitrogen application at later stages may also increase protein, calcium, and iron concentrations in tubers [20]. Therefore, the study of N application is an important aspect of potato production. In this study, we selected a medium-to-late maturing potato variety, Chuan-Yu 50, and set different N application rates and base/topdressing ratios to investigate the interaction of N fertilizer dosage and topdressing ratios on potato growth and development by measuring agronomic traits, photosynthetic traits, tuber quality, and yield at different stages. The aim was to find the best ratio of N fertilizer to reduce N application and increase the efficiency of potato production.

## 2. Materials and Methods

### 2.1. Site Description and Materials

The experiment was carried out on 21 December 2018 at the experimental site of Wenjiang, Sichuan, China (30°42′57.80″ N, 103°52′25.65″ E). in a humid subtropical climate with an average temperature of 11.91 °C, 196.9 mm average precipitation, and 11.53 h average sunshine time in growing seasons. Soil texture here is silt loam, with 29% sand fraction, 50% silt, and 21% clay fraction in the topsoil, which contains soil chemical properties at the 0–20 cm layer of 1.43 g kg$^{-1}$ total N, organic matter 23.29 g kg$^{-1}$, pH value 5.33, available potassium 84.00 mg kg$^{-1}$, available phosphorus 26.99 mg kg$^{-1}$, ammonium nitrogen 10.32 mg kg$^{-1}$, and nitrate nitrogen 18.85 mg kg$^{-1}$. The test material was Chuan-Yu 50, provided by the Sichuan Academy of Agricultural Sciences (Chengdu, China), medium to late maturity varieties, having a whole growth period of 85 d, sowed when the ground temperature exceeds 7 °C. Urea (N 46%) was used as nitrogen fertilizer, calcium superphosphate ($P_2O_5$ 12%) for phosphorus fertilizer, and potassium sulfate ($K_2O$ 50%) for potassium fertilizer.

### 2.2. Experimental Design and Management

In this study, randomized block trials were conducted. Three N levels were applied in the main treatment: N75 (75 kg ha$^{-1}$), N150 (150 kg ha$^{-1}$), N225 (225 kg ha$^{-1}$), and the N fertilizer base/topdressing ratio treatments were T1 (2:8), T2 (5:5), T3 (8:2), and T4 (10:0). The ratio represents the base fertilizer dosage/topdressing dosage of N. Twelve cross combinations were placed in 36 blocks (three replications) randomly. Topdressing was performed at the beginning of potato bud emergence (37 days after breeding) and N fertilizer was applied between two adjacent potato plants in the holes. The experiment was carried out as ridge culture, planting with a ridge height of 20 cm, row spacing of 17 cm × 70 cm, and plot area of 11.2 m$^2$. Phosphorus and potassium fertilizers were applied at 60 kg ha$^{-1}$ and 300 kg ha$^{-1}$ levels, respectively, when sowing. Pre-emergence

herbicides were sprayed after sowing, followed by black mulching, and the fertilizer dosage at different stages of each treatment is shown in Table 1.

**Table 1.** Fertilizer application amount and nitrogen fertilizer based/topdressing ratio (kg ha$^{-1}$).

| Nitrogen Input | Treatment | Nitrogenous Fertilizer | | | | Phosphate Fertilizer | Potassic Fertilizer |
|---|---|---|---|---|---|---|---|
| | | T1 | T2 | T3 | T4 | | |
| N75 | B | 15 | 37.5 | 60 | 75 | 60 | 300 |
| | T | 60 | 37.5 | 15 | 0 | 0 | 0 |
| N150 | B | 30 | 75 | 120 | 150 | 60 | 300 |
| | T | 120 | 75 | 30 | 0 | 0 | 0 |
| N225 | B | 45 | 112.5 | 180 | 225 | 60 | 300 |
| | T | 180 | 112.5 | 45 | 0 | 0 | 0 |

Note: B—Base N amount, T—Topdressing N amount.

### 2.3. Plant Sampling and Measurements

2.3.1. Agronomic Traits

Representative plants under healthy conditions and free of pests and diseases were taken at the budding stage (10 d after topdressing) and the tuber maturation stage. Plant height was calculated as the length from the stem base to the highest point of the plant, and stem thickness was measured using vernier calipers as the average transverse/longitudinal width at the stem base. Twenty fresh leaves were selected and punched 3 times per leaf with a hole punch (inner diameter = 0.5 mm), and the ratio between the fresh weight of the punched leaves and the fresh weight of the whole plant was calculated as the total area of the punched leaves over the leaf area of the single plant, and the leaf area index (LAI) = total leaf area/plant area [21]. The root system was scanned using a scanner (Epson V700 Photo, Seiko Epson Corporation, Beijing, China) and analyzed for total root length, root surface area, root mean diameter, root volume, and root tip number using the WinRHIZO PRO 2013 system software [22] (Regent Inc., Quebec City, QC, Canada).

2.3.2. Photosynthesis, Substance Metabolism, and Accumulation

Traits were measured in the budding stage (10 d after topdressing) and the tuber maturation stage. The net photosynthetic rate (Pn), intercellular $CO_2$ concentration (Ci), transpiration rate (Tr), and stomatal conductance (Cond) of functional leaves (inverted 3–4 leaves) were measured using a portable photosynthesis system (Li-6400, LI-COR Inc., Lincoln, NE, USA) at 9:00–11:00 a.m. on a sunny day. SPAD values were measured using a chlorophyll rapid tester (SPAD-502 PLUS, Konica Minolta Inc., Tokyo, Japan). All determination indices were measured on three individual plants per plot with three replications. Samples were classified into paper bags according to root, stem, leaf, and tuber organs, removed green at 105 °C for 30 min, dried at 85 °C constant weight, and then weighed for dry matter. The N content of different parts of the plant was measured using a Kjeldahl nitrogen meter (UDK169, VELP Inc., Milan, Italy) by digesting samples with concentrated $H_2SO_4$ and $H_2O_2$ [23]. Leaf-soluble sugar content was assessed using an anthrone-sulphuric acid colorimetric [24], and leaf-soluble protein content was stained with Komas Brilliant Blue G-250 [25]. Starch content in the tubers was measured using potato starch as the standard, OD606-OD415 for amylose and OD715-OD539 for amylopectin [26]. Crude protein content was calculated by conversion coefficient 6.25 to N [27,28]. Ascorbic acid content was assessed by its ability to reduce TCA-iron ion solution [29].

2.3.3. Yield Component and Yield

Harvested on 1 June 2019, when half of the plant leaves in the whole field turned yellow, excluding diseased, rotten, and insect-infested potatoes. To calculate the average yield (kg ha$^{-1}$), we counted the number of effective plants in each plot firstly, then weighed and counted the yield and number after harvesting all of them. The number of tubers per plant, average single potato weight, and commercial rate were counted by 10 plants per plot.

The commercial potato rate was classified as commercial potatoes according to potatoes that are more than 50 g, with smooth surface and healthy flesh [30]. In general, 1.5% of the total weight of the harvested potatoes was deducted as impurities and soil content.

### 2.4. Date Analysis

Variance analysis was performed using IBM SPSS Statistics 23 one-way variance analysis (ANOVA) to determine the level of significant effect of N fertilizer used and base/topdressing ratio at 0.01 and 0.05 levels. Significance tests between means under each treatment were performed using LSD with $p < 0.05$. Image mapping, data fitting, and PCA analysis were performed using Origin 2020. Fitting the yield to the amount of nitrogen applied the correlation coefficient *r* was calculated by Spearman's rank correlation coefficient.

## 3. Results

### 3.1. Agronomic Traits

There was a highly significant ($p < 0.01$) positive reciprocal effect of N use and the base/topdressing ratio on agronomic traits at bud emergence and tuber maturity stage. N dosage was highly significant ($p < 0.01$) affecting potato plant height and stem thickness up to maturity after based application, while the base/topdressing ratio showed a weaker effect on plants' agronomic traits (Figure 1). The leaf area index of T4 treatments averaged 8.55%, 14.3%, and 15.0% higher during the maturation period (Figure 1B). Though the trend was T3 > T2 > T1 > T4 at the budding stage, similar trends can be found at plant height, stem diameter, and root length. The root systems of the potato plants were analyzed at the budding stage, 10 days after topdressing. The appropriate base fertilizer concentration was found to be beneficial to potato root establishment and total root length increased with the increase of base N fertilizer (Figure 1D), and there was a highly significant interaction ($p < 0.01$) between base and topdressing N. Among them, the N150T3 fertilization ratio of the root surface area, root mean diameter, root volume, and root tip number was significantly greater than other treatments (Figure 1E–H).

### 3.2. Leaf Photosynthetic Traits and Metabolites

The overall photosynthetic capacity of the leaf in the budding stage was strongest with higher SPAD and Pn at 225 kg ha$^{-1}$ (Figure 2A–C), while low base N fertilization resulted in a decrease in leaf Pn (Figure 2A). Cond, Ci, and Tr were positively correlated with the amount of N applied (Figure 2B,C). However topdressing was beneficial in improving the photosynthetic capacity of the potato leaves, the effect of fertilizer supplementation was not satisfactory with insufficient base fertilizer (Figure 2D–F), and the high N topdressing led to a decrease in leaf SPAD, Pn, Cond, Ci, and Tr, which were observed at 10 days of topdressing at bud emergence (Figure 2A–C). At the tuber maturation stage, Pn was generally T2 > T3 > T1 > T4, and N75T2, N150T4, N225T2 occupied the maximum radar area respectively, while fertilization mainly affected leaf Tr and Cond. Leaf soluble protein content was positively correlated with the amount of N fertilizer available to the plant, while soluble sugar content showed a negative trend (Figure 3), so that fertilization at bud emergence may not immediately affect leaf metabolism.

### 3.3. N Content and Dry Matter of Different Organs

There was a highly significant ($p < 0.01$) positive interaction between N application and base/topdressing ratio on potato plant dry matter accumulation, which was manifested at tuber maturity (Figure 2). With the increase of the nitrogen application rate, the dry matter accumulation of plants first increased and then decreased The amount of base fertilizer mainly affected the dry matter accumulation in potato leaves, the N150T3 and N150T4 leaves' dry matter was higher than other treatments at the budding stage. Moreover, T3 under different N levels reaches the highest. At the stage of tuber maturation, the N150 treatment had the highest plant and tuber dry matter; T3 tuber dry matter accumulation was significantly higher ($p < 0.05$) than the other treatments, averaging 4.77–15.7%.

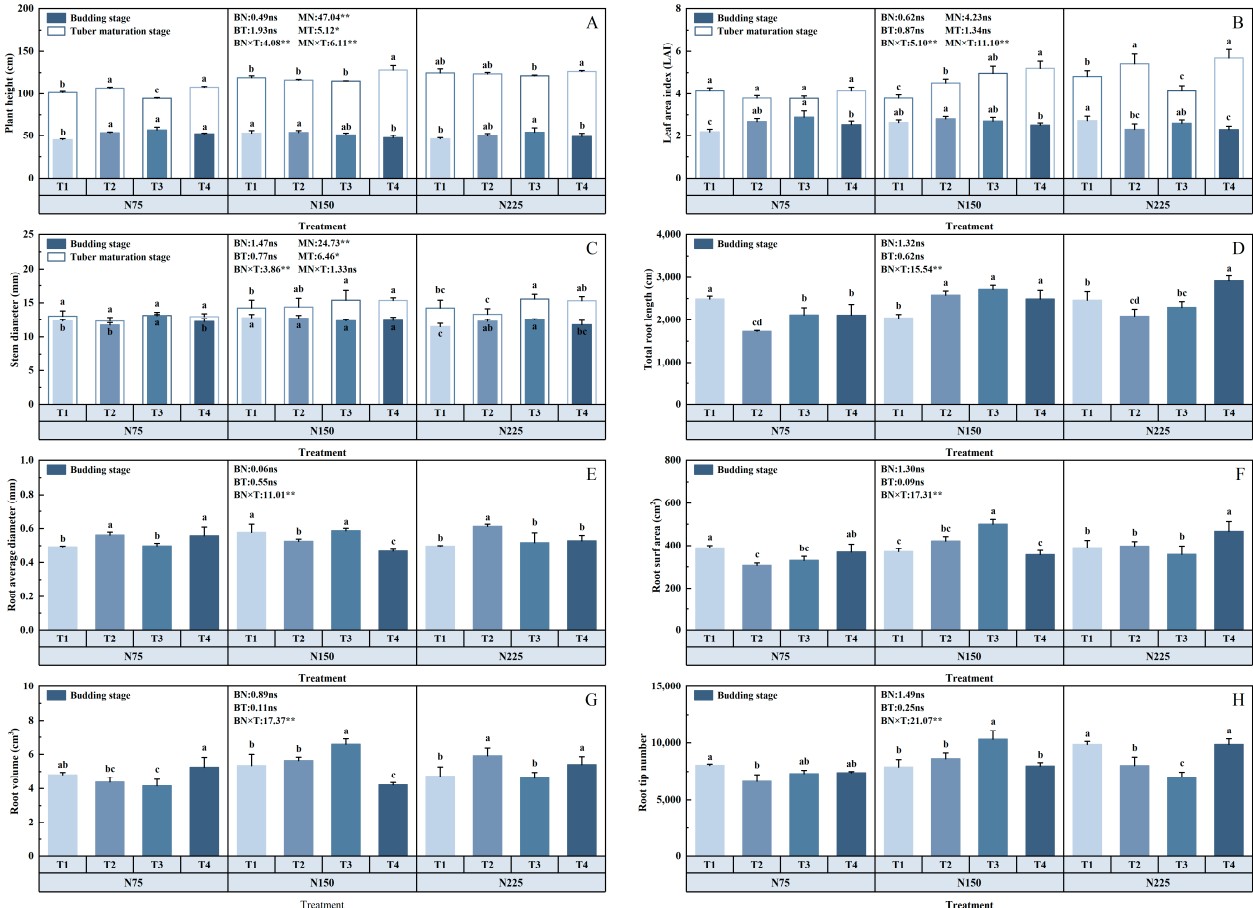

**Figure 1.** Agronomic characteristics of potatoes in different periods, plant height (**A**), LAI (**B**) and stem diameter (**C**) overground. Underground part of roots, total root length (**D**), root average diameter (**E**), root surf area (**F**), root volume (**G**) and root tip (**H**). Solid blocks represent the mean value of the budding stage and hollow blocks for the tuber maturation stage. Capital letter before F-value: B, budding stage; M, tuber maturation stage; N, nitrogen levels; T, base/topdressing ratio, the values refer to the F-value of ANOVA. Bars indicate standard errors (*n* = 3), and different lowercase letters indicate significant differences between treatments (*p* < 0.05). ns, not significant, * *p* < 0.05, ** *p* < 0.01.

The nitrogen uptake and utilization characteristics of potatoes differed at different breeding periods, and as potatoes grew the nitrogen content of the plant decreased to synthesize other metabolites. Total plant nitrogen accumulation in N75, N150, and N225 treatments was 1.33 g, 1.45 g, 1.41 g at the budding stage, 13.81 g, 16.38 g, and 13.82 g at tuber maturity, and the highest in the N150 treatment. N225T2 had a significantly (*p* < 0.05) higher N content than the other treatments, while T1 had the lowest percentage, showing a trend of T3 > T4 > T2 > T1 overall. Although supplemental N fertilization had little effect on the average N content in the tubers (Figure 4D), tubers' dry matter accounted for 68.65–82.10% of the plant dry matter content at the tuber maturity stage (Table 2).

### 3.4. Tuber Quality, Yield, and Yield Components

The crude protein and ascorbic acid content of tubers showed an increasing trend with increasing N application (Figure 5A), and there was a highly significant (*p* < 0.01) positive interaction between N application and b/topdressing ratio. The crude protein content of T1 under N225 was significantly (*p* < 0.05) higher than that of N75 and N175 by 21.8% and 6.92%, respectively. The ascorbic acid content of N225T2 was significantly (*p* < 0.05) higher than that of the other treatments. Topdressing also had an effect on tuber starch content (Figure 5B), and there was a tendency for tuber starch to decrease at harvest with

increasing fertilizer application, and the groups with higher amylose/amylopectin ratios were N75T1 > N225T4 > N150T4 > N225T2.

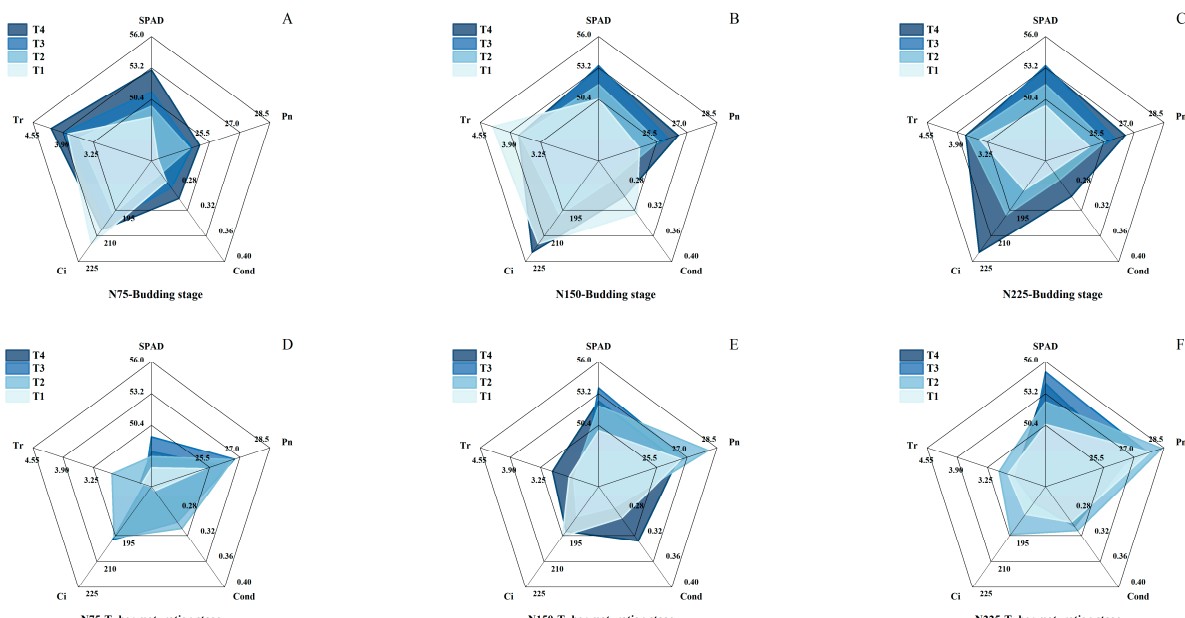

**Figure 2.** Comparison of photosynthetic characters of potato leaf at budding stage (**A**–**C**) and tuber maturation stage (**D**–**F**). Each angle of the pentagon showed different traits, SPAD, Pn-net photosynthetic rate ($\mu$mol $CO_2$ m$^{-2}$ s$^{-1}$), Cond-stomatal conductance (mol $H_2O$ m$^{-2}$ s$^{-1}$), Ci-Intercellular $CO_2$ concentration ($\mu$mol $CO_2$ mol$^{-1}$), Tr,-transpiration rate (mmol $H_2O$ m$^{-2}$ s$^{-1}$). Base/topdressing ratio of T1, T2, T3, and T4 is 2:8, 5:5, 8:2 and 10:0, respectively.

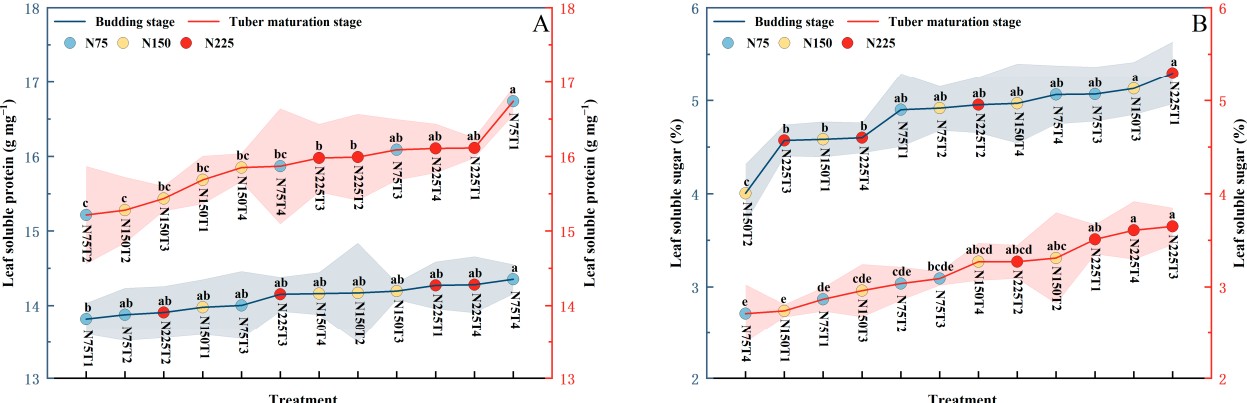

**Figure 3.** Leaf soluble protein (**A**) and leaf soluble sugar (**B**) content, broken lines show means of the budding stage (blue) and tuber maturation stage (red), shade along the broken line indicate standard errors (*n* = 3), different lowercase letters indicate significant differences between treatments (*p* < 0.05). Balls of N75 in blue, N150 in yellow, and N225 in red are sorted by size.

With N usage increased, the average single potato weight decreased (Table 3). N225 was significantly (*p* < 0.05) lower than N150 by 4.19%, while the number of potatoes per plant increased, N225 was higher than N150 and N75 by 3.33% and 7.26%, respectively, at *p* < 0.05 levels. The average yield under N150 was the highest, significantly (*p* < 0.05) higher than N75 and N225 by 2143.85 kg ha$^{-1}$ and 2016.61 kg ha$^{-1}$, respectively. T3 and T4 had the best performance under different base/topdressing ratios and different N application rates. The commercial potato rate of T3 and T4 under N150 treatment was higher than other treatments at the *p* < 0.05 level, with T3 yielding the highest. Overall, the interaction between N application and the base/topdressing ratio on yield composition

was not significant, however, N application was highly significant ($p < 0.01$) and positively correlated with the number of potatoes set per plant, and the base/topdressing ratio was highly significant ($p < 0.01$) and positively correlated with yield.

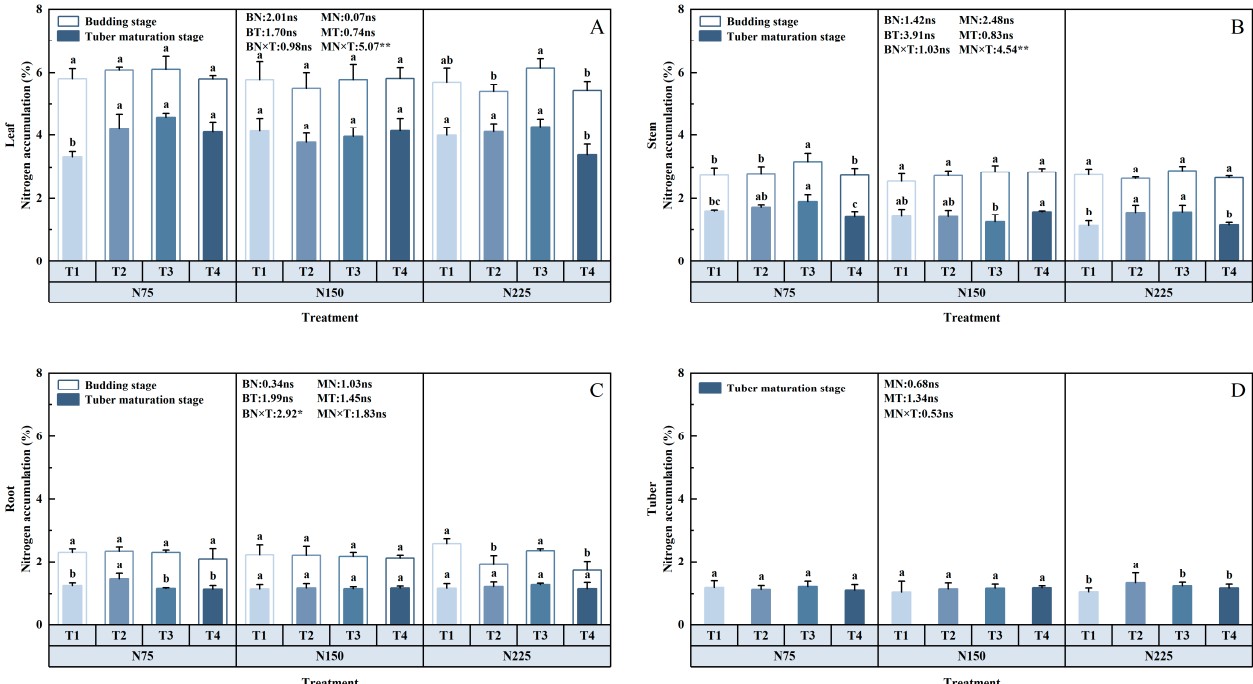

**Figure 4.** Nitrogen accumulation in leaf (**A**), stem (**B**), root (**C**), and tuber (**D**) at the budding stage (hollow block) and tuber maturation stage (solid block). Capital letter before F-value: B, budding stage; M, tuber maturation stage; N, nitrogen levels; T, base-topdressing ratio. Bars indicate standard errors (*n* = 3), and different lowercase letters indicate significant differences between treatments ($p < 0.05$). Ns, not significant, * $p < 0.05$, ** $p < 0.01$.

**Table 2.** Effects of different nitrogen fertilizer treatments on potato dry matter accumulation in different stages (g).

| Stage | | Budding Stage | | | | Tuber Maturation Stage | | | | |
|---|---|---|---|---|---|---|---|---|---|---|
| Treatment | | Root | Stem | Leaf | Plant | Root | Stem | Leaf | Tuber | Plant |
| N75 | T1 | 1.24 a | 3.36 b | 6.18 b | 10.78 b | 2.99 a | 18.96 ab | 18.96 a | 127.39 a | 168.31 a |
| | T2 | 1.01 b | 4.13 a | 6.21 b | 11.35 b | 2.59 ab | 18.00 b | 16.26 b | 133.60 a | 170.45 a |
| | T3 | 0.91 b | 4.35 a | 7.67 a | 12.93 a | 2.33 b | 13.50 c | 13.61 c | 135.05 a | 164.50 a |
| | T4 | 1.17 a | 4.12 a | 7.32 a | 12.62 a | 2.57 b | 19.33 a | 16.07 d | 133.34 a | 171.42 a |
| | Average | 1.08 b | 3.99 b | 6.85 c | 11.92 b | 2.62 b | 17.44 c | 16.23 c | 132.35 b | 168.67 c |
| N150 | T1 | 1.16 a | 5.07 a | 7.69 b | 13.92 a | 2.68 c | 19.99 d | 17.51 c | 127.25 c | 167.45 c |
| | T2 | 1.40 b | 4.25 b | 7.30 b | 12.94 b | 3.20 ab | 24.01 c | 19.99 b | 157.02 b | 204.23 b |
| | T3 | 1.20 a | 4.60 ab | 8.34 a | 14.14 a | 3.49 a | 30.52 b | 23.50 a | 171.92 a | 229.43 a |
| | T4 | 1.16 b | 4.50 b | 8.44 a | 14.09 a | 2.98 bc | 36.40 a | 24.05 a | 149.41 b | 212.85 b |
| | Average | 1.23 a | 4.61 a | 7.94 a | 13.77 a | 3.09 a | 27.73 a | 21.27 a | 151.40 a | 203.49 a |
| N225 | T1 | 1.27 a | 5.12 a | 7.61 ab | 13.99 a | 3.06 a | 28.45 a | 22.71 a | 118.72 b | 172.95 b |
| | T2 | 0.95 b | 4.72 ab | 8.15 a | 13.82 a | 2.67 a | 21.90 d | 18.08 b | 129.52 ab | 172.17 b |
| | T3 | 0.95 b | 4.90 ab | 7.61 ab | 13.46 ab | 2.89 a | 24.58 c | 18.07 b | 136.00 a | 181.54 ab |
| | T4 | 0.93 b | 4.51 b | 7.17 b | 12.60 b | 2.92 a | 26.61 b | 22.84 a | 139.09 a | 191.46 a |
| | Average | 1.02 b | 4.81 a | 7.64 b | 13.47 a | 2.89 a | 25.39 b | 20.42 b | 130.83 b | 179.53 b |
| ANOVA | N | 2.14 | 4.41 | 3.70 | 5.68 * | 2.10 | 5.11 | 3.56 | 6.94 * | 5.74 * |
| | T | 1.02 | 0.26 | 1.02 | 0.51 | 0.07 | 0.96 | 0.64 | 3.79 | 1.51 |
| | N × T | 8.65 ** | 4.79 ** | 9.5 ** | 7.23 ** | 5.11 ** | 116.04 ** | 68.44 ** | 4.35 ** | 12.04 ** |

Data are the means of three replicates. Different letters in rows represent significant ($p < 0.05$) differences between different base/topdressing ratios, F-values of N, T, and N × T have degrees of freedom at 2, 3, and 6, respectively. * $p < 0.05$, ** $p < 0.01$.

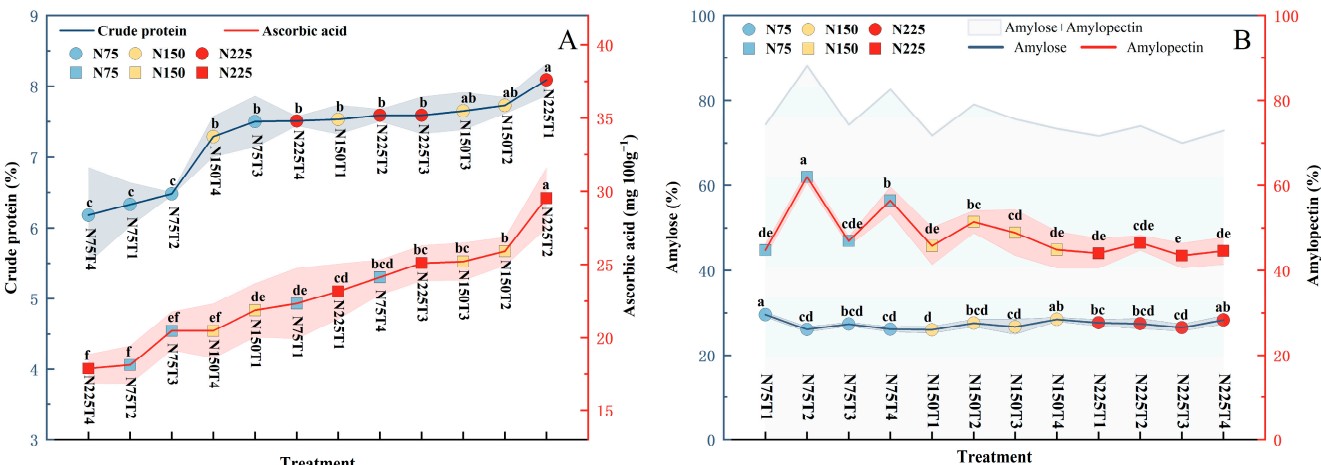

**Figure 5.** Tuber crude protein (**A** blue) and ascorbic acid (**A** red) content, tuber amylose (**B** blue) and amylopectin (**B** red) content, amylose added to amylopectin Broken lines show mean value, shade along the broken line indicates standard errors (*n* = 3), different lowercase letters. The results of the study show that the mean value of N75 in blue was higher than that of N75 in blue. Balls of N75 in blue, N150 in yellow, and N225 in red are sorted by size in (**A**), but by N management in (**B**).

**Table 3.** Effects of different treatments on potato yield components and yield.

| Treatment | | Average Tuber Weight (g) | Tuber Number per Plant | Commodity Potato Rate (%) | Average Yield (kg ha$^{-1}$) |
|---|---|---|---|---|---|
| N75 | T1 | 86.19 b | 6.07 c | 81.11 ab | 43,917.58 bc |
| | T2 | 88.43 b | 6.37 ab | 77.67 c | 47,289.04 a |
| | T3 | 92.09 ab | 6.50 a | 83.00 a | 50,237.17 a |
| | T4 | 98.3 a | 6.10 bc | 79.67 bc | 50,386.01 a |
| | Average | 91.25 a | 6.26 c | 80.42 a | 47,957.45 b |
| N150 | T1 | 83.64 b | 6.47 a | 79.00 b | 45,441.75 b |
| | T2 | 91.66 a | 6.63 a | 76.00 b | 51,108.46 a |
| | T3 | 96.67 a | 6.53 a | 83.33 a | 53,072.10 a |
| | T4 | 92.55 a | 6.53 a | 83.33 a | 50,782.84 a |
| | Average | 91.13 a | 6.54 b | 80.42 a | 50,101.30 a |
| N225 | T1 | 81.02 b | 6.63 b | 78.67 a | 45,126.68 b |
| | T2 | 80.24 b | 6.83 ab | 80.67 a | 46,070.78 b |
| | T3 | 88.47 a | 6.93 a | 81.67 a | 51,564.81 a |
| | T4 | 89.43 a | 6.60 b | 79.00 a | 49,576.50 b |
| | Average | 84.79 b | 6.75 a | 80.00 a | 48,084.69 b |
| ANOVA | N | 6.89 * | 26.22 ** | 0.05 | 4.17 |
| | T | 8.21 * | 5.99 * | 2.39 | 18.92 ** |
| | N × T | 1.67 | 0.93 | 3.62 * | 1.06 |

Data are means of three replicates. Different letters in rows represent significant (*p* < 0.05) differences between various base/topdressing ratio F-value of N, T, and N × T has a degree of freedom at 2, 3, and 6 respectively. * *p* < 0.05, ** *p* < 0.01.

### 3.5. Fitting Analysis and PCA Analysis

Fitting the yield to the amount of nitrogen applied (Figure 6) showed a highly significant (*p* < 0.05) positive correlation between the yield, total N application (Figure 6A), and topdressing N (Figure 6C). The amount of base N fertilizer applied was not significantly correlated with the yield (Figure 6B). The results of a PCA analysis at the budding stage explained 54.2% of the variance between the two axes (Figure 7A). The yield was mainly positively correlated with leaf SPAD, LAI, leaf net photosynthetic rate, plant dry matter, and leaf soluble protein content during this period. Results of a PCA at tuber maturity showed a total contribution of 51.8% variance explained by both axes (Figure 7B). When the growth center changed to underground growth, leaf soluble sugar content and plant

height, the LAI and SPAD values were positively correlated with yield. Plant N and leaf SP showed a positive correlation with Pn, but a negative correlation with tuber yield. At tuber maturity, the N75 treatment was more concentrated, while N150 and N225 treatments were more concentrated in a plane, indicating that base N fertilizer dose largely determined the condition of plant growth and the proportion of topdressing fertilizer should be coordinated with the N fertilizer dose.

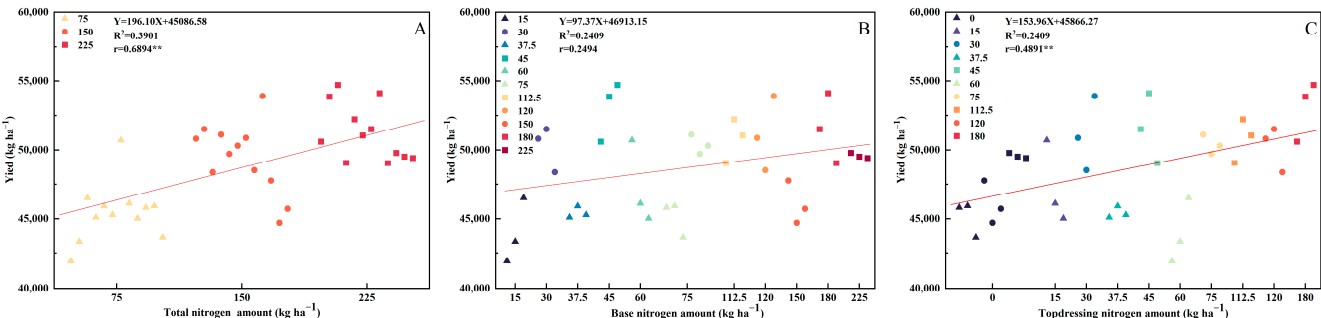

**Figure 6.** Fitting analysis of different nitrogen application rates and yields, total N amount (**A**), base N amount (**B**), topdressing N amount (**C**). $R^2$—linear correlation coefficient, r—Spearman rank correlation coefficient, ** $p < 0.01$.

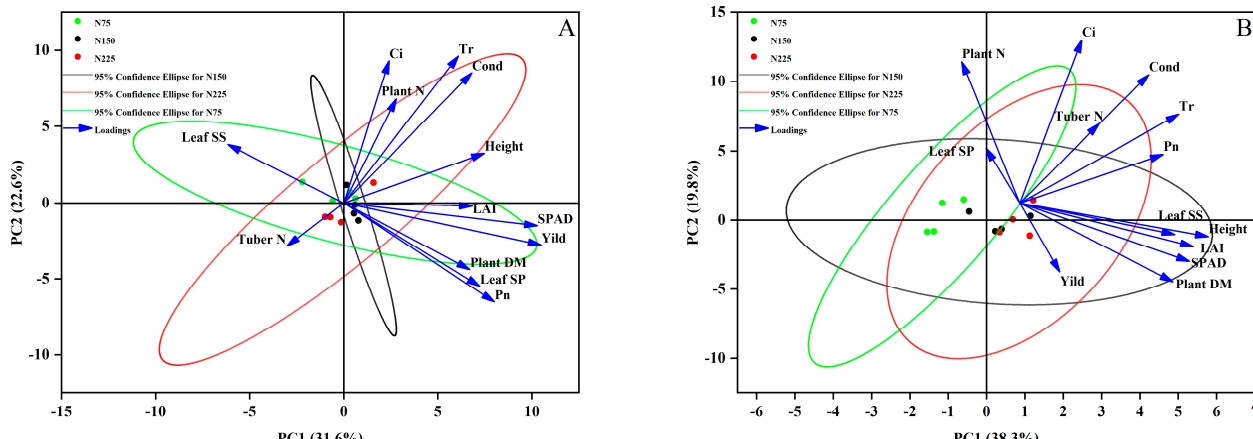

**Figure 7.** PCA analysis of traits at the budding stage (**A**) and tuber maturation stage (**B**). Leaf SS-leaf soluble sugar, leaf SP-leaf soluble protein, plant DM-plant dry matter, plant N-N in whole plant except tuber, tuber N-N in the tuber, Pn-net photosynthetic rate, Ci-intercellular $CO_2$ concentration, Cond-stomatal conductance, Tr-transpiration rate, LAI-leaf area index.

## 4. Discussion

As an essential element for potatoes, nitrogen application directly affects plant growth and development, for the growth of tubers is directly related to the plant's allocation of N. N application affects the plant's nitrate uptake process for transport to the leaves, including reduction and redistribution, thus effecting tuber yield formation [31]. Therefore, plant biomass and leaf metabolic capacity can reflect potato productivity [32].

The potato root system differs from the grass family in that its roots are distributed only in the shallow soil layer, which can compensate for the plant's nitrogen with maximum efficiency when fertilization is carried out at the peak of nitrogen demand [33]. This study found that the N application level and base/topdressing ratio had a significant impact on plant N requirements, both on above- and belowground plant growth, though base fertilizer mainly affected root growth, with the total root length, average root diameter, root surface area, root volume, and root tip number all at a high level of 150 kg ha$^{-1}$. N application at the budding stage, though aboveground differences were not significant, the leaf area index was affected at the later stage of topdressing. As the largest source of potatoes, leaf

carbohydrate metabolism was closely related to N nutrition [34], Xiong et al. [35] found that leaf physiological characteristics played a greater role than anatomical structure. Leaf SPAD and soluble protein and sugars showed a certain increasing trend with the increase of N topdressing, though the effect on Pn was negative at the high N topdressing ratio, and highest only at the appropriate total N interval. Leaf Cond, Cir, and Tr were increased under high N topdressing ratios. This result indicates that N fertilization can improve the metabolic capacity of potato leaves and increase photosynthetic capacity. In conclusion, increased leaf metabolic capacity for a period of time after N fertilization may be an important reason for improved yield.

Although N is an important element for plant growth, a high N supply is not conducive to tuber formation and dry matter accumulation [36]. Only by effectively regulating the internal metabolic effects of N on the plant and increasing the dry matter accumulation of tubers will it be beneficial for production [37]. In this experiment, the N content of both tubers and plants at the budding stage and tuber maturity were negatively correlated with the yields and dry matter content of plants. The maximum nitrogen uptake rate in potatoes reached about 60 days after seedling emergence [38]. This was also an important period for the rapid increase in the dry matter [39]. This experiment showed that based application of N could significantly affect the dry matter accumulation of the plant, however, the critical value was reached at about 112.5 kg ha$^{-1}$ N applied. The dry matter and N content of the plant decreased when the N application level increased to 225 kg ha$^{-1}$; especially in the tubers, the negative effect was significant. It has been suggested that the effect of N application on tuber yield depends on the harvest time [40]. The yield was determined by the ability of plant leaves to intercept radiation and convert it into dry matter below ground.

Libby R. et al. [41] found that applying more than 112 kg ha$^{-1}$ of N at seedling emergence and 56 kg ha$^{-1}$ at tuber initiation did not improve tuber yield and quality. The same results were obtained in this study, increasing total N application increased tuber protein and ascorbic acid content, while topdressing could increase tuber ascorbic acid content to some extent. The amylose/amylopectin ratio is an important index of potato processing, for a higher ratio leads to higher dextrinization difficulty, but the proportion of amylose decreases as the potato grows [42]. At low N levels, the higher the proportion of topdressing N applied, the higher the amylose/amylopectin ratio, while the opposite is found at high N. Amylopectin decreases with increasing nitrogen application and is highest at a 5:5 base/topdressing ratio. When the amount of base N is low, it can be compensated by topdressing during flowering, but when the amount of base N is sufficient, chasing will cause a decrease in average single potato weight and an increase in the number of potatoes set per plant, thus reducing the rate of commercial potatoes. When the amount of base fertilizer is insufficient, chasing can compensate for part of the loss, but still cannot offset the negative effect of N deficiency in the early stage. In this study, the average single potato weight and commercial potato yield were highest at 150 kg ha$^{-1}$, The high N level led to an increase in potato numbers per plant but reduced commercial potato yields. When base N was applied at 120–180 kg ha$^{-1}$ and the base/topdressing ratio was 8:2, the equivalent yield per hectare was higher, and the yield component configuration was better than the 10:0 ratio.

So far, nitrogen fertilizer was found to perform better in two split applications for sand loam [43], and topdressing makes lower N input in silt loam [44]. Potato cultivars are sensitive to different climates and soil types. In this study, the results indicate that the appropriate N application and topdressing ratio can effectively improve the commercial potato rate and yield in silt loam, while adequate N fertilizer supply in the early stage is the key factor to large potato formation.

## 5. Conclusions

Potatoes are at peak demand of nitrogen in the early budding stage, therefore proper fertilization at this time can supple plant N nutritional needs timely. By adjusting the amount of N fertilizer and the base/topdressing ratio, potato plant growth and development can be enhanced. Meanwhile, biological yield, photosynthetic characteristics,

plant nitrogen accumulation, and tuber quality and yield can be improved. The highest biological and tuber yields can be obtained when applying N fertilizer in the range of 120–180 kg ha$^{-1}$, and the ratio of 8:2 can also effectively improve the tuber quality and commercial potato yield in silt loam. However, too much N fertilizer can reduce yields and single potato weight by increasing the tuber numbers per plant. But the specific N input dosage is different in varieties of soil types and cultivars.

**Author Contributions:** Conceptualization, P.L. and S.Z.; Data curation, X.F. and Z.X.; Formal analysis, X.F. and Z.X.; Funding acquisition, Q.W. and S.Z.; Investigation, X.F. and Z.X.; Methodology, X.F., Z.X. and H.M.; Project administration, Q.W., P.L. and S.Z.; Supervision, P.L. and S.Z.; Writing—original draft, X.F. and F.W.; Writing—review & editing, X.F., Z.X., H.M. and F.W. All authors have read and agreed to the published version of the manuscript.

**Funding:** This work was supported by the Tackling Key Problems and Supporting Projects of Breeding in the Sichuan Province (Grant No.: 2021YFYZ0019; 2021YFYZ0005); Natural Science Foundation of Sichuan Province (2022NSFSC0014) and the Sichuan Potato Innovation Team (Grant No.: sccxtd-2023-09).

**Institutional Review Board Statement:** Not applicable.

**Data Availability Statement:** For additional information contact the author by correspondence.

**Acknowledgments:** Thanks to Crop Research Institute of Sichuan Academy of Agricultural Sciences for providing seed potatoes.

**Conflicts of Interest:** The authors declare no conflict of interest.

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
