# Peer review of "Effect of N Fertilizer Dosage and Base/Topdressing Ratio on Potato Growth Characteristics and Yield"

_agronomy, doi:10.3390/agronomy13030909_

Round 1

Reviewer 1 Report

In experimental Design and Management section and results section, the the analysis results of experimental design method should be descriptive clearly, such as the block effect of "randomized block design" that can not be found in Table 2 & 3. Moreover, in the results of ANOVA table in Table 2 & 3, the interaction effects are not always existed in each response variables, but the multiple comparison results for the interaction effect were shown for each response variables.

Author Response

a Word is submitted.

Dear reviewer,

Thank you for your thoughtful review and professional advice. Based on your suggestions, we have made corrected modifications on the revised manuscript. The language style and description of grammar are amended, and a language services is concerned if needed, we would like to show the details as follows:

Point 1: In experimental Design and Management section and results section, the the analysis results of experimental design method should be descriptive clearly, such as the block effect of "randomized block design" that can not be found in Table 2 & 3.

Response 1: Thanks the reviewer for the constructive comments. We have revised the introductory part according to your comments. Random block design is emphasized in experimental design and the NOTES of Fig.& Table for more clear description.

In this study, randomized block trials were conducted, three N levels were ap-plied in the main treatment: N75 (75 kg ha−1), N150 (150 kg ha−1), N225 (225 kg ha−1), and N fertilizer base/topdressing ratio treatments were T1 (2:8), T2 (5:5), T3 (8:2), T4 (10:0), the ratio represents the base fertilizer dose/topdressing dose. Twelve cross combinations were placed in 36 blocks (three replications) randomly. Topdressing is performed at the beginning of potato bud emergence(37 days after breeding) and N fertilizer applied between two adjacent potato plants in holes. The experiment was carried out as ridge culture, planting with a ridge height of 20 cm, row spacing of 17 cm × 70 cm, plot area of 11.2 m2. Phosphorus and potassium fertilizers were applied at 60 kg ha−1 and 300 kg ha−1 levels respectively when sowing. Line 90-98.

Point 2: Moreover, in the results of ANOVA table in Table 2 & 3, the interaction effects are not always existed in each response variables, but the multiple comparison results for the interaction effect were shown for each response variables.

Response 2: This study was carried out as a randomized block trials, so that dates are analyzed as a randomized block design and the F-values shown referred to random-effect model results. After analyzing, I found that the efficiency of randomized blocks were almost insignificant, so those dates were not presented on tables, BUT, there maybe some misunderstanding for me with the role of efficiency of randomized blocks, so I add them on table 2&3 as attachment of the response. If this part of analyzed results should be shown in manuscript, please inform me, and thank you for your kindly correction.

For the reason why “the interaction effects are not always existed in each response variables, but the multiple comparison results for the interaction effect were shown for each response variables.”, here I table the random-effect model and fixed -effect model F-value in follow table, it shows that results of FIXED-EFFECT having a much more highly significant interaction between treatments, but consider of the field test might involves complex error, so the random-effect results are showing in manuscript. Moreover I reinspect the dates and output analyzed dating, there could be some interaction between based N and topdressing amount, because the treatments of topdressing are carried out by separated the total usage of N fertilizer, so the N×T interaction may represent the N potato plants could obtain from soil to some extant. So the effects are not always existed in each response variables but in interaction. And the traits we observed are mainly about plant growing physiology, plant agronomic traits and dry matters for instance.

Reviewer 2 Report

Current topic.

Adequate title of paper.

Abstract clear and concise.

The introduction provide enough basic information to explain the aim of the research.

Results are clearly presented and adequately discussed.

The manuscript requires the following minor improvements:
1) in section Abstract (Line 20) and Materials and Methods (Line 76) specify the type of soil (according FAO classification) on which the experiment was set up.
2) The results of such research depend among other things, on the characteristics of the slimate and type of soil. This should be stated at the end of the discussion.

Reviewer 3 Report

On lines 36 and 37 what year was 359.1 million tons produced?

Line 42 “upto about” is not grammatically correct.

Lines 42-44 be specific and give the range in number from the cited articles.

Section 2.1 It is unclear when you state T4 treatment was the best to what you are comparing the T4 with.  Is this T4 under all three N levels or something else.  You do this for both the height, stem and root length data.

This is repeats again in section 2.3 when talking about dry matter.

Figure 3 might be better by comparing the treatments either at the same N level as a bar graph rather than a line graph as it is hard to see if the effect is the topdressing versus basal or the overall level of N.  Same for Figure 5 as some of the protein content data shows a dramatic difference at 75 for three of the four treatments but the 150 and 225 more or less the same.

Line 292 do you mean 120 or 150?
